# Soil Factors Key to 3,4-Dimethylpyrazole Phosphate (DMPP) Efficacy: EC and SOC Dominate over Biotic Influences

**DOI:** 10.3390/microorganisms12091787

**Published:** 2024-08-29

**Authors:** Tikun Guan, Jilin Lei, Qianyi Fan, Rui Liu

**Affiliations:** Beijing Key Laboratory of Farmyard Soil Pollution Prevention-Control and Remediation, College of Resources and Environmental Sciences, China Agricultural University, Beijing 100193, China; gtk961103@cau.edu.cn (T.G.); jllei@cau.edu.cn (J.L.); fffanqqqianyi123@163.com (Q.F.)

**Keywords:** 3,4-dimethylpyrazole phosphate, inhibition effects, abiotic factors, biotic factors, N_2_O emissions

## Abstract

Nitrification inhibitors like 3,4-dimethylpyrazole phosphate (DMPP) are crucial in agriculture to reduce nitrogen losses. However, the efficacy of DMPP varies in different soils. This microcosm incubation study with six soils was conducted to elucidate how soil abiotic factors (physicochemical properties) and biotic factors (nitrogen-cycling microbial abundance and diversity) influence the performance of DMPP. The DMPP efficacy was evaluated through the ammonium-N retention rate (NH_4_^+^_RA), inhibition rate of net nitrification rate (NNR_IR), and reduction rate of N_2_O emissions (N_2_O_ERR). The results showed that DMPP had significantly different effects on mineral nitrogen conversion and N_2_O emissions from different soils. NH_4_^+^_RA, NNR_IR, and N_2_O_ERR ranged from −71.15% to 65.37%, 18.77% to 70.23%, and 7.93% to 82.51%, respectively. Correlation analyses and random forest revealed abiotic factors, particularly soil EC and SOC, as the primary determinants of DMPP efficiency compared to microbial diversity. This study sheds new light on the complex interactions between DMPP efficacy and soil environments. The identification of soil EC and SOC as the dominant factors influencing DMPP efficacy provides valuable insights for optimizing its application strategies in agricultural systems. Future research could explore the mechanisms underlying these interactions and develop tailored DMPP formulations that are responsive to specific soil conditions.

## 1. Introduction

Nitrogen (N) is a crucial nutrient for plant growth and agricultural productivity. Synthetic N fertilizers, most commonly urea, have the highest production cost in China’s agricultural systems. Unfortunately, the nitrogen use efficiency (NUE) is low, with only approximately 40% of the applied N being taken up by growing crops. The rest is lost through ammonia (NH_3_) volatilization, nitrate (NO_3_^−^) leaching, and nitrous oxide (N_2_O) emissions [1]. These reactive N losses contribute to the detrimental eutrophication of surface waters, groundwater contamination, and increased greenhouse gas emissions, exacerbating climate change. Moreover, the challenge of ensuring food security is further compounded by climate change, which places additional stress on the agricultural systems. Annual crop yields need to be nearly 40% higher by 2050 [2]. Enhancing NUE and crop productivity in a sustainable manner has thus become a paramount societal challenge.

A prevalent strategy to elevate NUE and minimize N losses involves incorporating nitrification inhibitors into fertilizers. These inhibitors are chemical substances specifically formulated to delay the microbial oxidation of ammonium nitrogen (NH_4_^+^) to nitrite (NO_2_^−^) and NO_3_^−^ [3,4]. Among the available inhibitors, 3,4-dimethylpyrazole phosphate (DMPP) is noted for its efficiency, as it chelates copper ions in ammonia monooxygenase (AMO), reducing N_2_O emissions and enhancing NUE with a minimal application rate [5,6,7]. However, the efficacy of DMPP is often highly unpredictable, with its performance lasting for only a few weeks or even days [8]. Unfortunately, and for reasons largely unexplained, the effectiveness of DMPP varies significantly across different soil types [9].

Previous research has highlighted that the efficacy of DMPP varies significantly with the soil microbial community structure [10]. Variations in microbial community structure, particularly nitrifying and denitrifying microbes, are suggested to play a pivotal role in modulating DMPP efficacy [11]. Zhou et al. [10] found that DMPP was more effective in reducing N_2_O production in soils where ammonia-oxidizing bacteria (AOB) rather than ammonia-oxidizing archaea (AOA) were the primary nitrifying microbe. Further evidence comes from the study by Dong et al. [12], which notably showed that the inhibitory effect of DMPP on AOB was significantly stronger compared to AOA. This might be attributable to the diminished ability of DMPP to enrich species with high ammonia-oxidizing capacities within the AOB community after urea application. However, Liu et al. [13] reported that the DMPP application led to a significant decrease in ammonium oxidation rates while simultaneously increasing the abundance of AOB over AOA. Zhang et al. [14] also noted that the diversity of AOB increased with prolonged DMPP usage. Therefore, it is crucial to establish correlations between the soil biotic factors and the inhibitory effect of DMPP to uncover the underlying mechanism of its inconsistent efficacy.

The inconsistent inhibitory effects of DMPP have been attributed to a range of edaphic and environmental variables, including soil temperature, moisture, texture, pH, organic matter (SOM), organic carbon (SOC), and salinity [15]. These factors directly impact the persistence and performance of DMPP in the soil [16,17,18,19,20,21]. Organic matter content, particularly SOC, is a significant determinant of DMPP efficacy. As a heterocyclic compound, DMPP is easily adsorbed by SOC, reducing its availability in the soil [22,23,24]. Soil salinity also plays an essential role in DMPP efficacy, with studies showing that DMPP significantly reduced N_2_O emissions in low-salinity soils compared to non-saline soils [21]. Thus, a comprehensive analysis that combines the impacts of soil abiotic factors (soil physicochemical properties) and biotic factors (soil microbial abundance and diversity) on DMPP’s effectiveness, along with an assessment of their relative importance, is crucial for understanding the reasons behind the inconsistent inhibitory effects of DMPP.

This present microcosm incubation study aimed to explore the key factors that influence DMPP efficacy across six agricultural soils with varying edaphic properties. Specifically, we hypothesized that abiotic factors have a more significant effect on DMPP efficacy than biotic factors. To test this hypothesis, the relative contributions of these factors to the observed variable DMPP efficacy were quantified based on the NH_4_^+^ retention rate, the net nitrification rate, and N_2_O emissions. This research provides new insights into the relationship between soil properties and DMPP efficacy, informing strategies to improve the efficiency and sustainability of nitrification inhibitors in agricultural systems.

## 2. Materials and Methods

### 2.1. Field Sites and Soil Sampling

This experiment was carried out on a diverse array of soils collected from six sites across China, including Tianshui, Gansu (TS, 34°27′ N, 105°58′ E), Shihezi, Xinjiang (SHZ, 44°21′ N, 86°04′ E), Heze, Shandong (HZ, 34°34′ N, 115°31′ E), Daxing, Beijing (DX, 39°43′ N, 116°33′ E), Siping, Jilin (SP, 43°10′ N, 124°22′ E), and Zhuzhou, Hunan (HN, 27°50′14″ N, 113°24′48″ E). These selected sites encompass various land-use types (cropping, horticulture, and vegetable cultivation) and climatic conditions, which span different regions (warm temperate, temperate, and subtropical) across China. Six sites did not use DMPP during planting during the season.

In 2019, soil samples were meticulously gathered from a depth of 0–20 cm at each sampling location. Subsequently, the samples were transported to the laboratory, where extraneous matter such as plant residues, small stones, and contaminants were removed, and the soil was sieved through a 2 mm mesh. The soil was then divided into two portions: the first was air-dried for the assessment of soil physicochemical properties and for the execution of incubation experiments, while the second was stored at −80 °C for subsequent microbial analysis. Each treatment in the present experiment was replicated three times. The specifics of the sampling sites, including their coordinates and basic details, are depicted in Figure 1 and summarized in Table 1.

### 2.2. Soil Microcosm Incubation Experiment

Fifty grams of dry soil were placed in an incubation bottle that had a diameter of 10 cm and a height of 11 cm. Deionized water was evenly distributed over the soil in each bottle to adjust the soil moisture level to 65% of the water-filled pore space (WFPS) and then incubated at a temperature of 25 °C for five days. Following the pre-incubation period, two treatments were applied to each bottle: (1) NH_4_Cl and (2) NH_4_Cl + DMPP, with three replicates for each treatment. NH_4_Cl was applied at the rate of 100 mg N·kg^−1^, and DMPP was applied as 1% of the added NH_4_^+^-N applied. Throughout the incubation period, water was replenished every 2 days to maintain a constant soil moisture content.

### 2.3. Determination of Soil Mineral Nitrogen

The soil samples from the incubation bottles were destructively sampled on days 0, 1, 3, 7, and 10 for the analysis of changes in the concentrations of NH_4_^+^ and NO_3_^−^. The soils were extracted with a 1 M KCl solution in a 1:5 (*w*/*v*) ratio; subsequently, the concentrations of NH_4_^+^-N and NO_3_^−^-N in the extracts were determined using a flow analyzer (3-AA3 AutoAnalyzer, Bran Luebbe, Norderstedt, Germany) [16]. The calculations for the soil net ammonification rate (R_amo_) [25], NH_4_^+^-N retention rate (NH_4_^+^_RA), net nitrification rate (R_nit_) [25], and inhibition rate of the net nitrification rate (NNR_IR) are as follows:R_amo_ = (NH_4_^+^-N_i+1_ − NH_4_^+^-N_i_)/(t_i+1_ − t_i_)(1)
NH_4_^+^_RA = [(R_amo-CK_ − R_amo-T_)]/R_amo-CK_ × 100%(2)
R_nit_ = (NO_3_^−^-N_i+1_ − NO_3_^−^-N_i_)/(t_i+1_ − t_i_)(3)
NNR_IR = [(R_nit-CK_ − R_nit-T_)/R_nit-CK_] × 100%(4)

Note: NH_4_^+^-N_i_ and NO_3_^−^-N_i_ represent the soil NH_4_^+^-N and NO_3_^−^-N concentrations at the onset of incubation, respectively, whereas NH_4_^+^-N_i+1_ and NO_3_^−^-N_i+1_ represent the corresponding concentrations at the end of the incubation. Similarly, t_i_ and t_i+1_ signify the start and end of each incubation period, respectively. Moreover, R_amo-CK_ and R_nit-CK_ designate the net ammoniation rate and nitrification rate of the soil in the control group, whereas R_amo-T_ and R_nit-T_ represent the net ammoniation rate and nitrification rate of the soil in the experimental group, respectively.

### 2.4. Determination of N_2_O Emissions

On days 1, 3, 7, and 10, 20 mL of gas was collected from the headspace of each bottle using a 30 mL syringe. The gas was subsequently transferred to a 12 mL vacuum vial for the subsequent measurement of the N_2_O concentration. The N_2_O concentration was determined by gas chromatography (HP 6890, Agilent, Santa Clara, CA, USA), with peak areas automatically calculated by the chromatography software (EZChrom Elite, v3.3.2), thus enabling the determination of the N_2_O concentration [26,27]. The formulas for calculating the cumulative N_2_O emissions (N_2_O_ce_) and the reduction rate of the N_2_O emissions (N_2_O_ERR) were as follows:N_2_O_ce_ = ∑(F_i+1_ + F_i_)/2 × (T_i+1_ − T_i_) × 24(5)
N_2_O_ERR = [(N_2_O_ce-CK_ − N_2_O_ce-T_)/N_2_O_ce-CK_] × 100%(6)

Note: F_i_ and F_i+1_ represent the average emission fluxes of the gases collected at time points i and i + 1, respectively. The notation T_i_ − T_i+1_ refers to the time interval between the gas collections at i + 1 and i. Furthermore, N_2_O_ce-CK_ and N_2_O_ce-T_ signify the cumulative N_2_O emissions during the incubation process in the control group and the experimental group, respectively.

### 2.5. Soil DNA Extraction and High-Throughput Sequencing

Following the instructions of the E.Z.N.A.^®^ soil DNA kit (Omega Bio-tek, Norcross, GA, USA), total DNA extraction was performed on the background soil microbial communities. The quality of the DNA extraction was evaluated through 1% agarose gel electrophoresis, and the DNA concentration and purity were determined using a NanoDrop 2000 spectrophotometer (Thermo Fisher Scientific, Waltham, MA, USA). The samples were then stored at −20 °C until they were subjected to high-throughput sequencing analysis.

Polymerase chain reaction (PCR) amplification, targeting the AOA_*amoA* [28], AOB_*amoA* [29], *narG* [30], *nirS* [31], *nirK* [32], and *nosZ* [33] genes (the key genes involved in nitrification and denitrification), was conducted using the following program: initial denaturation at 95 °C for 3 min, followed by 27 cycles of denaturation at 95 °C for 30 s, annealing at 55 °C for 30 s, extension at 72 °C for 30 s, and a final extension at 72 °C for 10 min, concluding with a hold at 4 °C (PCR machine: ABI GeneAmp^®^ 9700, Thermo Fisher Scientific, Waltham, MA, USA). The PCR reaction mixture contained 4 μL of 5× TransStart FastPfu buffer, 2 μL of 2.5 mM dNTPs, 0.8 μL of each upstream and downstream primer (5 μM), 0.4 μL of TransStart FastPfu DNA polymerase, and 10 ng of template DNA, and was made up to a total volume of 20 μL. The amplification products were analyzed by 2% agarose gel electrophoresis and sequenced using the Illumina Miseq 300 sequencing platform (San Diego, CA, USA). Using fastp (v0.20.0) for quality assurance, FLASH (v1.2.7) for read assembly, and UPARSE (v7.1) for OTU clustering (97% similarity, chimera removal), the sequences were annotated with RDP classifier (v2.2) against Silva 16S rRNA (v138) at 70% confidence, revealing microbial diversity. The specific primers utilized are detailed in Appendix A. Each treatment in the present experiment was replicated three times.

### 2.6. Quantitative PCR of Functional Genes

Quantitative PCR (qPCR) was carried out on an ABI7300 real-time fluorescence quantitative PCR detection system, employing the ChamQ SYBR Color qPCR Master Mix (2×). The same primers used in the high-throughput sequencing were also utilized in this qPCR study. Before performing qPCR, we validated the specificity and efficiency of the primers used for each functional gene. This was conducted by performing qPCR with a range of template concentrations using standard curves generated from the known quantities of purified amplicons or cloned genes. The primers were deemed acceptable if they showed a high specificity (a single peak in the melting curve analysis) and amplification efficiency close to 100%. qPCR reactions were set up in triplicate for each sample to ensure reproducibility and minimize pipetting errors. Negative controls (no template) were included in every qPCR run to monitor for potential contamination.

The 20 μL amplification system was comprised as follows: 10 μL of 2× ChamQ SYBR Color Qpcr Master Mix, 0.8 μL of preprimers (5 μM), 0.4 μL of 50× ROX reference Dye 1, 2.0 μL of DNA sample, and 6.8 μL of sterilized water. Amplification was performed under the following conditions: an initial pre-denaturation step at 95 °C for 5 min, followed by a single repeat; denaturation at 95 °C for 30 s; annealing at 58 °C (for *nirK*, 55 °C was used) for 30 s, with this cycle repeated 35 times; and a final extension at 72 °C for 1 s. qPCR data were analyzed using the appropriate statistical methods to calculate the mean gene copy numbers and determine statistical significance. Any outliers or anomalous results were identified through a visual inspection of the amplification curves and were excluded from further analysis. The amplification efficiencies for the various genes were as follows: 102.44% for the AOA_*amoA* gene, 92.16% for the AOB_*amoA* gene, 105.92% for the *narG* gene, 98.72% for the *nirS* gene, 88.06% for the *nirK* gene, and 105.61% for the *nosZ* gene. Notably, the R² values for all genes exceeded 0.99.

### 2.7. Statistic Analysis

Data processing and the creation of tables were performed using Excel 2019, and statistical analyses were conducted with IBM SPSS Statistics 20 software. The results were used as the mean ± standard error (Mean ± SE). A one-way analysis of variance (ANOVA) with the least significant difference (LSD) method was employed to determine the significance of differences in the data (*p* < 0.05). Microbial diversity analysis was conducted using the Majorbio Cloud platform (https://cloud.majorbio.com. accessed on 18 March 2024) [34]. Correlation heatmap analysis was performed using the Tutu Cloud platform (https://www.cloudtutu.com. accessed on 20 March 2024). The random forest analysis (R software, v4.1.3) was conducted to quantify the relative importance of different soil physicochemical properties and microbial factors on DMPP efficacy. The variable importance scores were derived based on the reduction in the prediction error of the model when each variable was permuted out of the analysis. Finally, graphs were generated utilizing Origin 2024 software.

## 3. Results

### 3.1. The Physicochemical Properties of Different Soils

The physicochemical properties of soil varied significantly across different sites, as shown in Table 2. The pH of all the tested soils ranged between 4.44 to 7.99. The soil with the highest electrical conductivity (EC) was TS soil, recording a value of 326 μS·cm^−1^, followed by the DX and SHZ soils. Conversely, the EC values in the SP and HN soils were notably lower, at 71.73 and 101.53 μS·cm^−1^, respectively. The SOM and SOC contents in the HN soil were significantly higher than in the other five soils, with SOM at 58.56 g·kg^−1^ and SOC at 33.97 g·kg^−1^. The total nitrogen (TN) content in the soil followed the order of DX > HN > SP > SHZ > HZ > TS. Regarding the C/N ratio, it was ranked as SP > HN > SHZ > TS > DX > HZ. Additionally, the DX soil had the highest available phosphorus (AP, 123.64 mg·kg^−1^) and available potassium (AK, 404.18 mg·kg^−1^) levels.

### 3.2. Composition, Diversity, and Abundance of Soil Nitrogen-Cycling Microorganisms

The microbial community profiles, α-diversity index, and gene abundances associated with the nitrogen transformation in the tested soils are delineated in Figure 2 and Appendix A. Notably, there were pronounced differences in the composition of nitrogen-transforming microorganisms, especially ammonia-oxidizing microbes and denitrifiers, across the various soil types. In the HN soil, the α-diversity and gene abundances of AOA and AOB were the lowest compared to the other soils tested. Regarding gene abundances, the *narG*, *nirK*, and *nirS* genes were significantly more abundant in the soils of TS, SHZ, HZ, and DX compared to the SP and HN soils. However, the diversity index of the microorganisms involved in the denitrification process, specifically those harboring the *narG*, *nirK*, *nirS*, and *nosZ* genes, follow a consistent pattern across the six soil types.

Principal coordinate analysis (PCoA) was employed to provide a nuanced understanding of the disparities in the composition of nitrogen-transforming microbial communities across the different soils (Figure 3). The β-diversity analysis revealed statistically significant differences (*p* < 0.01) in the microbial community profiles of both ammonia-oxidizing and denitrifying microbes, specifically those carrying AOA_*amoA*, AOB_*amoA*, *narG*, *nirK*, *nirS*, and *nosZ* genes.

### 3.3. The Inhibitory Effects of DMPP on Nitrification and N_2_O Emissions from Different Soils

The impact of DMPP on nitrification inhibition is quantified through the alterations in the soil’s NH_4_^+^-N retention rate (NH_4_^+^_RA), the nitrification rate inhibition rate (NNR_IR), and the N_2_O emission reduction rate (N_2_O_ERR). The effectiveness of DMPP in inhibiting inorganic N transformations and reducing N_2_O emissions varies across different soil types, as outlined in Table 3. Following the addition of DMPP, the NH_4_^+^_RA was above 60% in the TS, HZ, and SP soils. Conversely, in the SHZ and DX soils, the NH_4_^+^_RA was approximately 27%. A negative NH_4_^+^_RA value of −71.15% was observed in the HN soil, indicating no inhibitory effect of DMPP and indicating ongoing nitrification processes. The NNR_IR in the TS and HZ soils reached up to 70%, while in the DX soil, this value was only 18.77%. The efficacy of DMPP in inhibiting N_2_O emissions, ranked from highest to lowest, was TS > SHZ > DX > HZ > HN > SP. Specifically, in the TS soil, DMPP inhibited 82.51% of the N_2_O production, whereas, in the HN soil, it only managed to inhibit 18.54% of the N_2_O emissions.

### 3.4. The Relative Contribution of Abiotic and Biotic Factors to DMPP Efficacy

The impact of soil abiotic factors (physicochemical characteristics) and biotic factors (α-diversity and the copy numbers of nitrogen-related microbial genes) on the effectiveness of DMPP is illustrated in Figure 4. A significant negative relationship exists between the SOM, SOC, and TN contents and NH_4_^+^_RA. Conversely, the Shannon index of the AOA_*amoA*, *nirK*, and *nosZ* genes, as well as the copy numbers of the AOA_*amoA* and *nirS* genes, show a significant positive correlation with NH_4_^+^_RA (Figure 4a). No significant correlation was observed between the soil physicochemical properties, α-diversity, and the copy numbers of nitrogen-related microbial genes and NNR_IR (Figure 4a).

Soil pH, EC, and the abundance of several genes involved in nitrogen transformation (including AOB_*amoA*, *narG*, *nirK*, *nirS*, *nosZ*) demonstrate a significant positive correlation with N_2_O_ERR, with the highest correlation coefficient reaching 0.92 (for the *nirK* gene copy number) (Figure 4a). Random forest analysis further confirms these correlations, with the soil SOC identified as the most significant contributor to NH_4_^+^_RA (IncNodePurity = 6577.8738), whereas the soil EC was found to be the most crucial factor affecting N_2_O_ERR (IncNodePurity = 2372.3203) (Figure 4b,c).

## 4. Discussion

### 4.1. The Abiotic and Biotic Properties and DMPP Efficacy Varied among Selected Soils

The choice of soil sampling sites, which encompass a diverse array of climatic zones in China, including warm temperate semi-humid and semi-arid climates, temperate continental climates, and subtropical monsoon humid climates, is illustrated in Figure 1 and detailed in Table 1. These sites demonstrate notable disparities in the annual average temperature, annual precipitation, crop types, and soil characteristics, reflecting the geographical and ecological complexities of China. Analyses of the soil physicochemical properties reveal substantial variations in the pH value, EC, SOM, SOC, TN, C/N, AP, and AK across the different regions, as shown in Table 2. For instance, TS and SHZ exhibit higher soil EC values, possibly due to salt accumulation resulting from the relatively dry climatic conditions in these regions [35]. DX and HN, on the other hand, show higher SOM and TN contents, which can be attributed to the relatively humid climate and abundant vegetation cover in these areas [36,37]. Moreover, HN has the lowest soil pH value, indicating acidic characteristics typical of subtropical soils [38].

Regarding microbial community diversity, an examination of the α-diversity indices and the copy numbers of specific functional genes (AOA_*amoA*, AOB_*amoA*, *narG*, *nirK*, *nirS*, and *nosZ*) (Appendix A) revealed significant differences in the background nitrogen-related microbial community structure among the various regions. DX and SP exhibited higher Shannon diversities, ACE, and Chao1 indices, indicating richer and more diverse microbial communities, possibly due to the higher moisture and nutrient levels in these regions under temperate monsoon climatic conditions [39,40]. Conversely, HN had a lower Shannon diversity but a higher Simpson index, which suggests a more dominant microbial community—a characteristic possibly attributable to the specific environmental conditions and soil physicochemical properties of the subtropical region [41]. These observations were consistent with previous studies, indicating that the structure and diversity of soil microbial communities vary significantly (Figure 2 and Figure 3, and Appendix A) and were influenced by a range of environmental factors, including the climate, soil type, vegetation cover, moisture, and nutrient status [42,43,44]. Consequently, the selection of these six soils served as a representative sample to investigate the soil factors that impact the efficacy of DMPP applications.

In the present investigation, DMPP demonstrated notable variations in its ability to inhibit mineral nitrogen transformation and N_2_O emissions—findings that aligned with earlier studies [45]. Specifically, DMPP notably enhanced the retention of NH_4_^+^ in the TS, SHZ, HZ, DX, and SP soils, while in the HN soil, DMPP did not significantly affect the concentrations of NH_4_^+^ and NO_3_^−^ and, instead, facilitated the loss and transformation of NH_4_^+^. Additionally, when subjected to the same N input, the N_2_O_ERR in the TS, SHZ, HZ, and DX soils augmented by DMPP was significantly higher compared to that in the SP and HN soils (Table 3). These discrepancies might be attributed to a multitude of factors, including the soil characteristics, the structure of the microbial community, and the environmental conditions. Nair et al. [17] scrutinized the efficacy of DMPP in differing soil types and concluded that the performance of DMPP is predominantly contingent upon the soil’s properties. Fan et al. [11] have highlighted that the differing niches of nitrogen-related microorganisms contribute to the varying effectiveness of DMPP across different soil types. Given these insights, it is imperative to explore the relative significance of both abiotic and biotic factors to optimize the application of DMPP for more effective nitrogen management.

### 4.2. The Physicochemical Properties of Soil Have a More Significant Impact on the Efficacy of DMPP

In the present research, the findings highlight that both abiotic and biotic factors exert a significant impact on the efficiency of DMPP (Figure 4a), underscoring the pivotal role of soil physicochemical properties, as well as microorganism biological characteristics, in determining the efficacy of DMPP [11,46]. Although the activities and diversity of microorganisms, which DMPP targets, play a crucial role in modulating its inhibitory capacity, the degradation and transformation of DMPP in soil are predominantly governed by the soil’s physicochemical properties. Prior research has shown that soil physicochemical properties are the primary determinants of DMPP’s effectiveness in mitigating N_2_O emissions, exerting a greater influence than the cumulative effect of microbial abundance and community structure [11].

The results unveiled that SOC and EC made the greatest contribution to NH_4_^+^_RA and N_2_O_ERR (Figure 4b,c), suggesting that under the experimental conditions, abiotic factors predominated over biotic factors in influencing the efficacy of DMPP application. This substantiated the hypothesis of this study. The concentration of SOC was demonstrated to be negatively correlated with the efficacy of DMPP, likely due to the physicochemical adsorption of DMPP onto the soil matrix [47]. Moreover, SOC serves as an accessible energy source for non-targeted heterotrophic microorganisms that metabolize DMPP, which was associated with a diminished capacity of DMPP to inhibit ammonium oxidation [48]. The aforementioned evidence collectively indicated that in soils with lower SOC contents, the addition of DMPP was observed to result in a higher retention rate of NH_4_^+^ compared to the soils with higher SOC contents.

In this research, soil EC was demonstrated to be yet another crucial factor that positively impacts the efficacy of DMPP in mitigating N_2_O emissions. On the one hand, increased EC values usually indicate a higher concentration of soluble ions in the soil solution, perhaps reducing the intermolecular interactions of DMPP molecules through electrostatic repulsion, thereby enhancing the effectiveness of DMPP [49,50]. Li et al. [21] provided evidence showing that, in soils with relatively high salinity, DMPP displayed a higher efficacy in N_2_O inhibition compared to non-saline soils. On the other hand, EC has a significant influence on the composition, abundance, activity, and diversity of soil ammonia-oxidizing microbial communities [51,52,53,54,55,56,57]. Research has shown that the relative abundance of soil ammonia oxidizers is positively correlated with EC [58]. Hence, within a specific range, the higher the EC value of the soil, the greater the background abundance of ammonia oxidizers (Table 2 and Appendix A), leading to a more pronounced reduction in N_2_O emissions upon the addition of DMPP. However, there are also studies that have observed a negative effect of EC on the relative abundance of soil ammonia oxidizers, with excessively high EC values potentially impairing the physiological and metabolic functions of these microorganisms [59,60,61]. Further investigations are warranted to reconcile these contradictory findings.

In conclusion, while our study emphasizes the importance of abiotic factors in determining DMPP’s efficacy, it is evident that these factors interact with biotic components in complex ways. Future research should explore the mechanisms underlying these interactions in greater depth, incorporating multi-omics approaches to provide a more comprehensive understanding of the soil microbial ecology and its response to DMPP application. By elucidating these intricate relationships, we can develop tailored DMPP formulations and application strategies that are responsive to specific soil conditions, ultimately enhancing their effectiveness and sustainability in agricultural systems.

## 5. Conclusions

The present results indicate that both abiotic and biotic factors influence the efficacy of DMPP to varying degrees, with soil physicochemical properties, such as EC and SOC, playing a predominant role. These findings underscore the importance of considering soil physicochemical properties in the development and optimization of nitrification inhibition strategies. The knowledge gained from this research can contribute to the development of more targeted and effective nitrogen management strategies, thereby supporting sustainable agricultural practices and environmental protection.

## Figures and Tables

**Figure 1 microorganisms-12-01787-f001:**
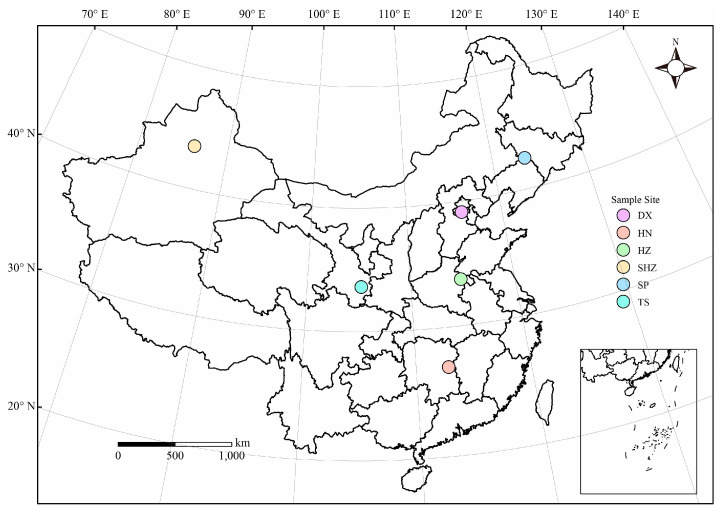
Distribution map of soil sampling points.

**Figure 2 microorganisms-12-01787-f002:**
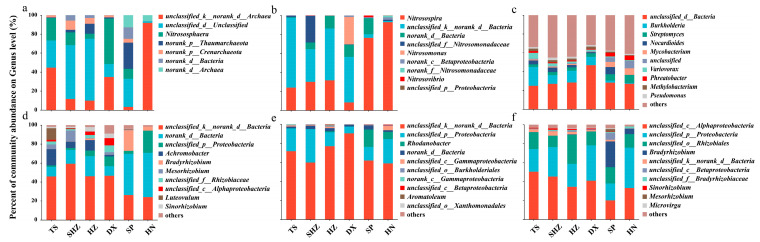
Genus-level microbial community composition. The relative abundance of the top 10 genera for each treatment in six soils is shown. ((**a**) AOA_*amoA*; (**b**) AOB_*amoA*; (**c**) *narG*; (**d**) *nirK*; (**e**) *nirS*; (**f**) *nosZ*).

**Figure 3 microorganisms-12-01787-f003:**
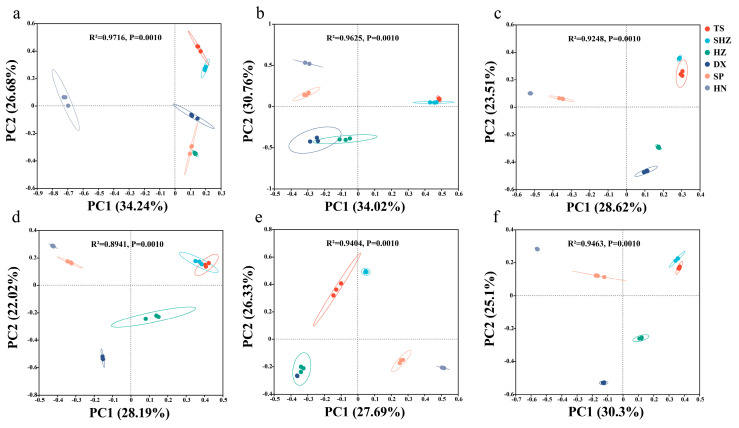
Differences in microbial community compositions (OTU level) related to soil nitrogen transformation were analyzed based on PCoA. Bray–Curtis algorithm was used for the distance between samples, and Adonis analysis was used to test differences between groups. ((**a**) AOA_*amoA*; (**b**) AOB_*amoA*; (**c**) *narG*; (**d**) *nirK*; (**e**) *nirS*; (**f**) *nosZ*).

**Figure 4 microorganisms-12-01787-f004:**
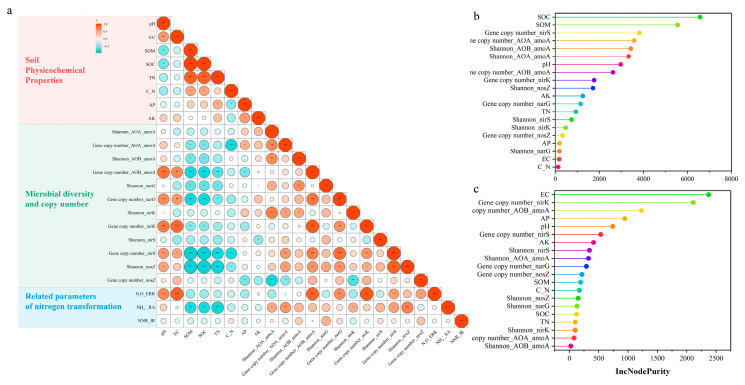
Correlation and random forest analysis of different soil physicochemical properties, microbial diversities, gene copy numbers, and related parameters of nitrogen transformation. (**a**) Correlation analysis was calculated using the Spearman algorithm; the size of the circular area represents the size of the correlation r, and the significance mark was *p* ≥ 0.05 without any mark; 0.01 < *p* < 0.05 *; 0.001 < *p* ≤ 0.05 **; *p* ≤ 0.001 ***. (**b**) The relative contribution of random forest analysis parameters to NH_4_^+^_RA; (**c**) the relative contribution of random forest analysis parameters to N_2_O_ERR.

**Table 1 microorganisms-12-01787-t001:** Basic information on soil sampling points.

Site	Number	Climate Type	Annual Mean Temperature (°C)	Mean Annual Precipitation (mm)	Grow Crops	Soil Type
Tianshui	TS	Warm temperate semi-humid semi-arid climate	11.04	612.23	Apple	Loessal soil
Shihezi	SHZ	Temperate continental climate	7.42	235.25	Cotton	Desert gray soil
Heze	HZ	Warm temperate semi-humid continental monsoon climate	15.35	583.43	Maize	Moisture soil
Daxing	DX	Warm temperate semi-humid continental monsoon climate	11.89	408.02	Pepper	Moisture soil
Siping	SP	Temperate monsoon climate	7.90	657.20	Maize	Chernozem
Hunan	HN	Subtropical monsoon humid climate	17.65	1286.30	Rice	Paddy soil

**Table 2 microorganisms-12-01787-t002:** Soil physicochemical properties.

Site	pH	EC (μS·cm^−1^)	SOM (g·kg^−1^)	SOC (g·kg^−1^)	TN (g·kg^−1^)	C/N	AP (mg·kg^−1^)	AK (mg·kg^−1^)
TS	7.88 ± 0.13 a ^#^	326.00 ± 75.51 a	15.34 ± 1.35 cd	8.90 ± 0.78 cd	0.85 ± 0.01 b	10.51 ± 0.80 ab	8.51 ± 0.69 e	29.17 ± 0.44 d
SHZ	7.79 ± 0.03 a	261.33 ± 15.45 a	20.35 ± 0.73 cd	11.80 ± 0.42 cd	1.04 ± 0.00 b	11.36 ± 0.40 a	15.31 ± 2.36 de	220.84 ± 3.08 b
HZ	7.99 ± 0.38 a	217.33 ± 2.40 ab	10.27 ± 0.61 d	5.96 ± 0.35 d	0.86 ± 0.12 b	6.93 ± 0.52 b	36.71 ± 1.94 b	163.88 ± 15.94 bc
DX	7.74 ± 0.04 a	264.67 ± 12.60 a	42.91 ± 3.44 b	24.89 ± 2.00 b	2.87 ± 0.26 a	9.12 ± 0.54 ab	123.64 ± 2.27 a	404.18 ± 33.48 a
SP	6.13 ± 0.09 b	71.73 ± 0.82 b	22.93 ± 0.89 c	13.30 ± 0.51 c	1.05 ± 0.08 b	12.79 ± 1.34 a	22.36 ± 0.54 cd	99.44 ± 7.61 cd
HN	4.44 ± 0.07 c	101.53 ± 1.43 b	58.56 ± 3.87 a	33.97 ± 2.25 a	2.72 ± 0.01 a	12.49 ± 0.82 a	27.68 ± 0.21 c	58.31 ± 2.08 d

Note: Results were used as mean ± standard error (mean ± SE), *n* = 3. ^#^ Different letters in the table indicate significant differences (*p* < 0.05). EC: electrical conductivity; SOM: soil organic matter; SOC: soil organic carbon; TN: total nitrogen; AP: available phosphorus; AK: available potassium.

**Table 3 microorganisms-12-01787-t003:** Effects of DMPP on soil NH_4_^+^-N retention, nitrification rate, and N_2_O emissions.

Soil Site	NH_4_^+^_RA (%)	NNR_IR (%)	N_2_O_ERR (%)
TS	65.37 ± 9.73 a ^#^	70.23 ± 2.08 a	82.51 ± 0.31 a
SHZ	27.17 ± 9.32 a	63.57 ± 1.99 ab	55.97 ± 0.02 b
HZ	61.27 ± 6.52 a	71.45 ± 0.96 a	28.91 ± 0.80 c
DX	27.01 ± 4.01 a	18.77 ± 2.92 b	51.30 ± 2.22 b
SP	63.43 ± 6.76 a	64.28 ± 1.69 ab	7.93 ± 0.88 e
HN	−71.15 ± 14.10 b	44.40 ± 23.46 ab	18.54 ± 0.63 d

Note: Results were used as mean ± standard error (mean ± SE), *n* = 3. ^#^ Different letters in the table indicate significant differences (*p* < 0.05). NH_4_^+^_RA: NH_4_^+^-N retention rate; NNR_IR: inhibition rate of net nitrification rate; N_2_O_ERR: reduction rate of N_2_O emission.

## Data Availability

The raw data supporting the conclusions of this article will be made available by the authors on request.

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
