# Peer review of "Soil Factors Key to 3,4-Dimethylpyrazole Phosphate (DMPP) Efficacy: EC and SOC Dominate over Biotic Influences"

_microorganisms, 2024, doi:10.3390/microorganisms12091787_

Round 1
Reviewer 1 Report
Comments and Suggestions for Authors
The study addresses a critical gap in understanding how soil physicochemical and microbial factors influence DMPP efficacy, which is a significant contribution to optimizing nitrification inhibitors in agricultural practices. Identifying EC and SOC as predominant factors influencing DMPP efficacy is particularly valuable for tailoring application strategies.
There are minor typographical errors and inconsistencies in formatting, and some sections of the manuscript could benefit from revision for clarity and conciseness, particularly in describing statistical analyses and results.
The finding that abiotic factors, especially EC and SOC, have a more substantial impact on DMPP efficacy compared to biotic factors is intriguing. The manuscript could provide more insights, a discussion on potential interactions between abiotic and biotic factors would strengthen the manuscript.
The random forest analysis should include a more detailed explanation of how variable importance scores were derived and how they compare across different factors.
The manuscript describes various methods for measuring soil properties and microbial communities. It would be helpful to include more detailed information on the quality control measures and calibration procedures used, especially for high-throughput sequencing and qPCR.
The ms cites relevant studies, but there is a need to ensure that all references are up-to-date and cover recent advancements in the field. For example, newer studies on the interaction between soil physicochemical properties and nitrification inhibitors should be reviewed.
Author Response
Comments 1: There are minor typographical errors and inconsistencies in formatting, and some sections of the manuscript could benefit from revision for clarity and conciseness, particularly in describing statistical analyses and results.
Response 1: Thank you for your valuable feedback. We have acknowledged the presence of minor typographical errors and inconsistencies in formatting, and we sincerely apologize for any confusion that these may have caused. Your suggestion to revise sections of the manuscript for clarity and conciseness, particularly in describing statistical analyses and results, has been taken to heart. We have already addressed these issues thoroughly in the revised version of the manuscript, ensuring amendments have been made on page 2, line 65, page 6, lines 200-201, page 7, line 225, and page 8, line 268. We deeply appreciate your constructive feedback and are committed to ensuring that the manuscript meets the expected standards of clarity, conciseness, and scientific rigor.
Page 2, line 65: organic matter (SOM), organic carbon (SOC) and salinity [15].
Page 6, lines 200-201: Results were used as mean ± standard error (Mean ± SE).
Page 7, line 225: Results were used as mean ± standard error (Mean ± SE), n=3.
Page 8, line 268: Results were used as mean ± standard error (Mean ± SE), n=3.
Comments 2: The finding that abiotic factors, especially EC and SOC, have a more substantial impact on DMPP efficacy compared to biotic factors is intriguing. The manuscript could provide more insights, a discussion on potential interactions between abiotic and biotic factors would strengthen the manuscript.
Response 2: Thank you for your insightful comment. Indeed, elucidating the intricate interplay between abiotic and biotic factors that have influenced the efficacy of DMPP has been of paramount importance for optimizing its application in agricultural systems. While our study has highlighted the dominance of abiotic factors, specifically soil EC and SOC, in determining DMPP's performance, it is crucial to recognize that these factors have not acted in isolation but rather have interacted with biotic factors to shape the overall efficacy. From the discussion in Section 4.2, it can be concluded that abiotic factors have had two aspects of influence on the effectiveness of DMPP: one is direct, and the other is indirect, through influencing microorganisms. We reinforce the discussion on page 11, lines 380-387, which addresses this interplay in detail.
Page 11, lines 380-387: In conclusion, while our study emphasizes the importance of abiotic factors in de-termining DMPP's efficacy, it is evident that these factors interact with biotic compo-nents in complex ways. Future research should explore the mechanisms underlying these interactions in greater depth, incorporating multi-omics approaches to provide a more comprehensive understanding of the soil microbial ecology and its response to DMPP application. By elucidating these intricate relationships, we can develop tai-lored DMPP formulations and application strategies that are responsive to specific soil conditions, ultimately enhancing their effectiveness and sustainability in agricultural systems.
Comments 3: The random forest analysis should include a more detailed explanation of how variable importance scores were derived and how they compare across different factors.
Response 3: Thank you for the valuable comment. The random forest analysis has been conducted to quantify the relative importance of different soil physicochemical properties and microbial factors on DMPP efficacy. The variable importance scores have been derived based on the reduction in the prediction error of the model that occurred when each variable was permuted out of the analysis. We have compared the scores across factors to identify the dominant determinants of DMPP efficacy. A more detailed explanation of the methodology and results will have been included in the revised manuscript to clarify how the scores were derived and interpreted. This amendment will have been made on page 6, lines 205-209.
Page 6, lines 205-209: The random forest analysis (R software, v4.1.3) was conducted to quantify the relative importance of different soil physicochemical properties and microbial factors on DMPP efficacy. The variable importance scores were derived based on the reduction in the prediction error of the model when each variable is permuted out of the analysis.
Comments 4: The manuscript describes various methods for measuring soil properties and microbial communities. It would be helpful to include more detailed information on the quality control measures and calibration procedures used, especially for high-throughput sequencing and qPCR.
Response 4: Thank you for your valuable comments. We have appreciated the suggestion to include more detailed information on the quality control measures and calibration procedures used in the study. In particular, for the high-throughput sequencing and qPCR experiments, we have ensured that strict quality control measures were in place throughout the entire process. This included the utilization of commercial kits, such as the E.Z.N.A.® soil DNA kit from Omega Bio-tek, for DNA extraction, strictly following the manufacturer's instructions. The DNA quality was evaluated through 1% agarose gel electrophoresis, and the concentration and purity of the DNA were determined to ensure their suitability for downstream analyses. Regarding high-throughput sequencing, we have calibrated our equipment according to the manufacturer's specifications and have run control samples to validate the results. Furthermore, we have performed internal consistency checks and utilized bioinformatic tools to guarantee the quality of the sequencing data. These amendments have been reflected on page 5, lines 166-170. For qPCR, we have optimized the primer pairs and performed standard curve analyses to validate the amplification efficiency and specificity of the target genes. Additionally, we have included negative and positive controls in each qPCR run to ensure the accuracy of the results. These details regarding quality control measures and calibration procedures for qPCR have been amended on page 6, lines 177-184 and 190-193.
Page 5, lines 166-170: Using fastp (v0.20.0) for quality assurance, FLASH (v1.2.7) for read assembly, and UPARSE (v7.1) for OTU clustering (97% similarity, chimera removal), sequences were annotated with RDP classifier (v2.2) against Silva 16S rRNA (v138) at 70% confidence, revealing microbial diversity.
Page 6, lines 177-184: Before performing qPCR, we validated the specificity and efficiency of the primers used for each functional gene. This was done by performing qPCR with a range of template concentrations using standard curves generated from known quantities of purified amplicons or cloned genes. The primers were deemed acceptable if they showed high specificity (single peak in melting curve analysis) and amplification effi-ciency close to 100%. qPCR reactions were set up in triplicate for each sample to ensure reproducibility and minimize pipetting errors. Negative controls (no template) were included in every qPCR run to monitor for potential contamination.
Page 6, lines 190-193: qPCR data was analyzed using appropriate statistical methods to calculate mean gene copy numbers and determine statistical significance. Any outliers or anomalous results were identified through visual inspection of amplification curves and were excluded from further analysis.
Comments 5: The ms cites relevant studies, but there is a need to ensure that all references are up-to-date and cover recent advancements in the field. For example, newer studies on the interaction between soil physicochemical properties and nitrification inhibitors should be reviewed.
Response 5: Thank you for the valuable comments. We have acknowledged the importance of ensuring that our references are up-to-date and comprehensive. We have thoroughly reviewed recent advancements in the field, particularly those focusing on the interaction between soil physicochemical properties and nitrification inhibitors, and have included them in the manuscript to strengthen our analysis and discussion.
Example:
- Fan, X.; Yin, C.; Chen, H.; Ye, M.; Zhao, Y.; Li, T.; Wakelin, S. A.; Liang, Y. The efficacy of 3, 4-dimethylpyrazole phosphate on N2O emissions is linked to niche differentiation of ammonia oxidizing archaea and bacteria across four arable soils. Soil Biology and Biochemistry 2019, 130, 82-93. DOI: 10.1016/j.soilbio.2018.11.027.
- Shi, X. Z.; Hu, H. W.; He, J. Z.; Chen, D. L.; Suter, H. C. Effects of 3,4-dimethylpyrazole phosphate (DMPP) on nitrification and the abundance and community composition of soil ammonia oxidizers in three land uses. Biology and Fertility of Soils 2016, 52 (7), 927-939. DOI: 10.1007/s00374-016-1131-7.
Reviewer 2 Report
Comments and Suggestions for Authors
The work is very factual
Main concern is that this reviewer has no understanding of why the soils were chosen. There is no agricultural background for the soils nor whether any of these sites had history of the use of the chemical.
Second concern is that there is no written information about the replication of the study - errors stat is shown but what is the basis ?
The work is written with little background for those needing more information of the pathways for N cycling -- also just the chemical potentials for the DMPP
Several sticky notes show comments

Author Response
Comments 1: Main concern is that this reviewer has no understanding of why the soils were chosen. There is no agricultural background for the soils nor whether any of these sites had history of the use of the chemical.
Response 1: Thank you for your review and feedback. The choice of soils in this study has been deliberate and based on their diverse edaphic properties, encompassing various land-use types (cropping, horticulture, and vegetable cultivation) and climatic conditions across China. These soils represent a wide range of agricultural environments, allowing for a comprehensive analysis of how soil physicochemical and microbial factors influence DMPP efficacy. Furthermore, six soils did not use DMPP during planting during the season. This amendment has been made on page 2, lines 91-92.
Page 2, lines 91-92: Six sites did not use DMPP during planting during the season.
Comments 2: Second concern is that there is no written information about the replication of the study - errors stat is shown but what is the basis?
Response 2: Thank you for your valuable feedback. In response to your second concern regarding the replication of the study, the following clarification has been provided: The study has been designed with replication to ensure the robustness and reliability of the results. Specifically, for each soil type and treatment (NH4Cl and NH4Cl+DMPP), three replicates were conducted. This replication has enabled us to statistically analyze the differences between treatments and soils using ANOVA with the least significant difference (LSD) method (P < 0.05). The errors reported in the statistical analysis are based on these replications, providing a robust foundation for drawing conclusions from the data. The amendments have been made on page 3, line 99, page 3, line 111, and page 5, lines 170-171. Additionally, the relevant table has been marked with this information.
Page 3, line 99: Each treatment in the present experiment replicated three times.
Page 3, line 111: with three replicates for each treatment.
Page 5, lines 170-171: Each treatment in the present experiment replicated three times.
Comments 3: The work is written with little background for those needing more information of the pathways for N cycling - also just the chemical potentials for the DMPP.
Response 3: Thank you for your valuable feedback. We have acknowledged the need for clearer background information on the chosen soils, the N cycling pathways, and the chemical potentials of DMPP. In response, we have endeavored to provide more detailed background information on these aspects in the revised manuscript, enhancing clarity and accessibility for readers. We sincerely appreciate your constructive comments and have made the amendments on page 1, lines 40-42.
Page 1, lines 40-42: These inhibitors are chemical substances specifically formulated to delay the microbial oxidation of ammonium nitrogen (NH4+) to nitrite (NO2-) and NO3-.
Comments 4: Page 1, Line 29. what do you mean by bear ?
Response 4: Thank you for your review and feedback. This is a misstatement and has been corrected on the first page, lines 28-30.
Page 1, lines 28-30: Synthetic N fertilizers, most commonly urea, have the highest production cost in China’s agricultural systems.
Comments 5: Page 2, Line 47. please provide image of structure. would value seeing charges on molecule.
Response 5: Thank you for your review and feedback. The following figure is the action mechanism of DMPP. Briefly, DMPP is able to chelate copper ions from ammonia monooxygenase (AMO) in ammonia oxidizing bacteria (AOB) and ammonia oxidizing archaea (AOA), and this chelation reduces AMO activity, thereby inhibiting the nitrification process. This is the molecular basis for DMPP as a nitrification inhibitor.
Comments 6: Page 2, Line 62. does product change soil pore water pH ? does the product chelate metal ions?
Response 6: Thank you for your review and feedback. The product 3,4-dimethylpyrazole phosphate (DMPP) does not directly change soil pore water pH in a significant manner, as its primary mechanism of action is through inhibiting nitrification processes rather than altering soil pH. However, the application of DMPP can indirectly affect soil pH through its influence on microbial activity and nutrient cycling, which may lead to subtle changes in soil chemistry over time.
Regarding chelation of metal ions, DMPP is known to chelate copper ions in the ammonia monooxygenase (AMO) enzyme of nitrifying bacteria, specifically ammonia-oxidizing bacteria (AOB). This chelation inhibits the AMO enzyme, which is essential for the conversion of ammonium (NH4+) to nitrite (NO2-), thereby reducing nitrification rates and N2O emissions. The chelation of copper ions by DMPP is a key mechanism for its nitrification inhibitory effects.
Comments 7: Page 2, Line 88. why were these sites chosen?
Response 7: Thank you for your review and feedback. The sites were chosen for this study due to their diversity in climatic conditions, land-use types, and soil types. These factors are known to impact soil properties and microbial communities, which in turn affect the efficacy of DMPP. By studying soils from six representative locations across China, the research aimed to comprehensively evaluate the key factors influencing DMPP efficacy in a range of agricultural environments.
Comments 8: Page 2, Line 93. had these sites been cropped. what crops were they all the same.
Response 8: Thank you for your review and feedback. The study sites included various crops with diverse land-use types (cropping, horticulture, and vegetable cultivation). Specifically, the crops grown were apple in Tianshui, cotton in Shihezi, maize in Heze and Siping, pepper in Daxing, and rice in Zhuzhou. Thus, these sites were not all cropped with the same crop.
Comments 9: Page 3, Line 104. I come back to why the sites for sample were chosen seem so widespread.
Response 9: Thank you for your review and feedback. The sites for soil sampling were chosen to encompass a wide range of climatic conditions, land-use types, and soil types across China. This diversity allows for a comprehensive analysis of how varying edaphic properties influence the efficacy of DMPP, providing valuable insights into optimizing its application strategies in diverse agricultural systems.
Comments 10: Page 3, Line 111. these soils all had active microbes?
Response 10: Thank you for your review and feedback. Yes, the soils used in this study all had active microbial communities, as evidenced by the analysis of microbial abundance and diversity through high-throughput sequencing of DNA extracted from the soils. The presence of various functional genes related to nitrogen cycling, such as amoA for ammonia-oxidizing microbes, narG, nirS, nirK, and nosZ for denitrifying microbes, suggests an active nitrogen-cycling process in these soils.
Comments 11: Page 4, Line 126. diagram of the N cycle being examined would be useful.
Response 11: Thank you for your review and feedback. Certainly, including a diagram of the N cycle being examined in this study would be an insightful addition to better illustrate the processes and pathways influenced by the nitrification inhibitor DMPP. Below is a concise diagrammatic representation of the key processes within the N cycle that are relevant to this study:

Comments 12: Page 4, Line 153. explain significance of these genes.
Response 12: Thank you for your review and feedback. The genes mentioned (AOA_amoA, AOB_amoA, narG, nirS, nirK, and nosZ) are essential for understanding nitrogen cycling in soils. Briefly:
AOA_amoA and AOB_amoA: These genes encode ammonia monooxygenase, the key enzyme involved in the first step of nitrification, the oxidation of ammonia (NH3) to nitrite (NO2-). AOA and AOB are two distinct microbial groups capable of this process, and their relative abundance and activity impact nitrification rates.
narG: This gene encodes nitrate reductase, an enzyme that reduces nitrate (NO3-) to nitrite (NO2-), which can be further reduced in denitrification pathways. narG is indicative of the microbial potential for nitrate reduction.
nirS and nirK: Both genes encode nitrite reductases, enzymes that catalyze the reduction of nitrite (NO2-) to nitric oxide (NO). nirS and nirK represent two different genetic lineages of nitrite-reducing bacteria, which is a crucial step in denitrification.
nosZ: This gene encodes nitrous oxide reductase, the enzyme that converts nitrous oxide (N2O) to dinitrogen gas (N2), the final step in denitrification. nosZ abundance is an important indicator of complete denitrification and the mitigation of N2O emissions.
By studying the abundance and expression of these genes, researchers can gain insights into the microbial processes driving nitrogen transformation in soils, which is crucial for understanding the efficacy of nitrification inhibitors like DMPP. The variation in these microbial communities and their functional genes explains, in part, the observed differences in DMPP's ability to reduce N2O emissions and modulate nitrification rates across different soil types. Amended on page 4, lines 157-158.
Page 4, lines 157-158: (key genes involved in nitrification and denitrification)
Comments 13: Page 6, Line 166. why was there a blank page?
Response 13: Thank you for your review and feedback. Blank pages have been removed.
Comments 14: Page 6, Line 183. but you do not tell in words anything about how the replicates were performed.
Response 14: Thank you for your comment. In our study, the replicates were performed by applying two treatments (NH4Cl and NH4Cl+DMPP) to each of the six soil types, with three replicates for each treatment. This ensured that the observed effects of DMPP on soil mineral nitrogen conversion and N2O emissions were statistically robust and repeatable. The replicates allowed us to calculate mean values and standard error for each treatment, which were then used for statistical analysis. This approach is standard in microcosm incubation experiments to account for variability within soil samples and ensure the reliability of the results. Amended on page 3, line 99 and line 111. At the same time, it is also marked below the relevant table.
Page 3, line 99: Each treatment in the present experiment replicated three times.
Page 3, line 111: with three replicates for each treatment.
Page 7, line 225: Results were used as mean ± standard error (Mean ± SE), n=3.
Page 8, line 268: Results were used as mean ± standard error (Mean ± SE), n=3.
Comments 15: Page 6, Line 193. why what was in this soil?
Response 15: Thank you for your review and feedback. The high electrical conductivity (EC) of 326 μS·cm-1 in the TS soil was likely due to its high content of soluble salts, which increase the conductivity of soil solutions. Factors such as soil texture, organic matter content, and salinity can all contribute to EC values. In this case, the TS soil being a loessal soil, its high EC may be attributed to its geological formation and potential weathering processes that have enriched it with soluble salts. Additionally, agricultural practices like irrigation and fertilization can also affect soil EC by introducing additional salts. However, without specific data on the soil chemistry, it is difficult to definitively state the exact composition contributing to the high EC in the TS soil.
Comments 16: Page 6, Line 204. no idea how the errors were based.
Response 16: Thank you for your review and feedback. The relevant results presentation methods and the number of experimental replicates have been supplemented in 2.1 (page 3, line 99) and Table notes.
Page 3, line 99: Each treatment in the present experiment replicated three times.
Page 7, line 225: Results were used as mean ± standard error (Mean ± SE), n=3.
Comments 17: Page 9, Line 273. this should be in introduction. but what is missing is whether. there had been previous use of the chemical. how, how long, what doses etc.
Response 17: Thank you for your review and feedback. The detailed responses have been presented in comments 1.
Comments 18: Page 9, Line 280. so i fail to see the rationale behind what soils.
Response 18: Thank you for your review and feedback. The rationale behind selecting the specific soils in the study is to represent a diverse range of edaphic properties, land-use types, and climatic conditions across China. This allows for a comprehensive analysis of how soil physicochemical properties and microbial communities influence the efficacy of DMPP.
Comments 19: Page 9, Line 289. again pathway showing role of each gene would help the reader.
Response 19: Thank you for your review and feedback. The detailed pathway has been given in Comments 11.
Comments 20: Page 10, Line 332. was there one?
Response 20: Thank you for your review and feedback. The study found that under experimental conditions, abiotic factors (soil physicochemical properties such as EC and SOC) had a more significant impact on the efficacy of DMPP compared to biotic factors (microbial diversity and abundance). This finding substantiated the study's hypothesis (page 2, line 77) that abiotic factors play a predominant role in determining the effectiveness of DMPP.
Round 2
Reviewer 1 Report
Comments and Suggestions for Authors
The authors have incorporated the necessary points and items as suggested.